# Face and Predictive Validity of MI-RAT (*M*ontreal *I*nduction of *R*at *A*rthritis *T*esting), a Surgical Model of Osteoarthritis Pain in Rodents Combined with Calibrated Exercise

**DOI:** 10.3390/ijms242216341

**Published:** 2023-11-15

**Authors:** Colombe Otis, Emilie Bouet, Sokhna Keita-Alassane, Marilyn Frezier, Aliénor Delsart, Martin Guillot, Agathe Bédard, Jean-Pierre Pelletier, Johanne Martel-Pelletier, Bertrand Lussier, Francis Beaudry, Eric Troncy

**Affiliations:** 1Groupe de Recherche en Pharmacologie Animale du Québec (GREPAQ), Department of Biomedical Sciences, Faculty of Veterinary Medicine, Université de Montréal, Saint-Hyacinthe, QC J2S 2M2, Canada; colombe.otis@umontreal.ca (C.O.); emilie.bouet@umontreal.ca (E.B.); ndeye.sokhna.keita@umontreal.ca (S.K.-A.); marilyn.frezier@umontreal.ca (M.F.); alienor.delsart@umontreal.ca (A.D.); martin.guillot@yahoo.ca (M.G.); bertrand.lussier@umontreal.ca (B.L.); francis.beaudry@umontreal.ca (F.B.); 2Charles River Laboratories Montreal ULC, Senneville, QC H9X 1C1, Canada; agathe.bedard@crl.com; 3Osteoarthritis Research Unit, Université de Montréal Hospital Research Center (CRCHUM), Montréal, QC H2X 0A9, Canada; dr@jppelletier.ca (J.-P.P.); jm@martelpelletier.ca (J.M.-P.); 4Centre de Recherche sur le Cerveau et L’Apprentissage (CIRCA), Université de Montréal, Montréal, QC H3T 1P1, Canada

**Keywords:** osteoarthritis, exercise, pain, nociception, animal model

## Abstract

Validating animal pain models is crucial to enhancing translational research and response to pharmacological treatment. This study investigated the effects of a calibrated slight exercise protocol alone or combined with multimodal analgesia on sensory sensitivity, neuroproteomics, and joint structural components in the MI-RAT model. Joint instability was induced surgically on day (D) 0 in female rats (*N* = 48) distributed into sedentary–placebo, exercise–placebo, sedentary–positive analgesic (PA), and exercise–PA groups. Daily analgesic treatment (D3–D56) included pregabalin and carprofen. Quantitative sensory testing was achieved temporally (D–1, D7, D21, D56), while cartilage alteration (modified Mankin’s score (mMs)) and targeted spinal pain neuropeptide were quantified upon sacrifice. Compared with the sedentary–placebo (presenting allodynia from D7), the exercise–placebo group showed an increase in sensitivity threshold (*p* < 0.04 on D7, D21, and D56). PA treatment was efficient on D56 (*p* = 0.001) and presented a synergic anti-allodynic effect with exercise from D21 to D56 (*p* < 0.0001). Histological assessment demonstrated a detrimental influence of exercise (mMs = 33.3%) compared with sedentary counterparts (mMs = 12.0%; *p* < 0.001), with more mature transformations. Spinal neuropeptide concentration was correlated with sensory sensitization and modulation sites (inflammation and endogenous inhibitory control) of the forced mobility effect. The surgical MI-RAT OA model coupled with calibrated slight exercise demonstrated face and predictive validity, an assurance of higher clinical translatability.

## 1. Introduction

Osteoarthritis (OA) is a primary cause of functional disability. It is characterized by joint pain, tenderness, and limitation in movement, leading to a quality-of-life loss, especially in women [1,2]. The etiology of OA is multifactorial and includes either an idiopathic cause and/or factors such as joint injury, obesity, age, and heredity [1]. This complex degenerative process affects all joint components. It is characterized locally by bone remodeling, synovium inflammation, cartilage loss, and the initiation of a complex nociplastic pain process [3,4]. At present, no cure for OA is available. The first-line treatments are limited to acetaminophen and nonsteroidal anti-inflammatory drugs, physical therapy, and lastly, joint replacement surgery when possible [5,6]. As the etiology of OA remains incompletely understood, various animal models have been developed to characterize the features of OA and pain mechanisms under different experimental conditions [7].

Rheumatologists’ positions on exercise in OA have been contradictory. On one hand, there was no significant enhancement or even alteration in cartilage structure [8], while on the other hand, it improved OA patients’ well-being [9,10]. Exercise is widely recommended in national and international OA management guidelines [5,6]. However, its effectiveness varies due to different factors, including the location of OA (most notable in knee OA) [9], the patient’s age and body condition, the nature of exercise [11], being maximal at around two months, and then vanishing, and being no better than usual care at 9–18 months in the case of hip and knee OA [10]. Similarly, in neuropathic human conditions, moderate aerobic exercise training combined with aerobic and resistance exercise training or high-intensity interval training reduced some, but not all, measures of neuropathic pain [12].

Experimental studies in rodents exhibit a similar ambivalent profile. Initially, in rat models of peripheral nerve injury, a meta-analysis of 14 studies [13] and a narrative review [12] substantiated a compelling counteracting impact of exercise on neuropathic hypersensitivity encompassing both mechanical withdrawal threshold and thermal withdrawal latency. In healthy rats, endogenous opioids [14], nitrergic [15], and noradrenergic [16] pathways are involved in exercise-induced antinociception. In neuropathic rat models, exercise-induced anti-hypersensitization was associated with neurotrophins [17], neurotransmitters, inflammatory actors, microglial activity, and opioid and endocannabinoid systems [12]. Globally, it is recognized that caloric restriction increases life expectancy in rodents, whereas exercise improves the quality of healthspan [18].

Interestingly, in mice eating a very high-fat diet, short-term (4 terminal weeks in a total of 24) moderate exercise reduced OA lesions in the medial femur induced by this alimentary regimen [19]. Exercise of moderate intensity on a treadmill in OA rat models exerted a beneficial influence, which was noted by reduced chondrocyte loss and various anti-inflammatory effects [20,21]. However, interval training did not prevent cartilage degeneration while improving subchondral bone turnover [22]. A dose–response relationship between exercise and lesion severity in an experimental surgically induced OA model is suggested, with slight (<7.5 km) and moderate (11–15 km) intensity exercises undertaken over two [23] to four [20,23] weeks on a treadmill being beneficial, but higher intensity exercise (30 km over four weeks) being nonbeneficial [23]. It has been shown that regular physical activity, specifically wheel running, can produce analgesia through central inhibitory control involving opioidergic and serotoninergic systems in a mouse model of chronic muscle hyperalgesia [24]. A recent narrative review highlighted that many studies investigating the role of exercise in rodent models of arthritis, including OA, focused on molecular and pathophysiological joint changes as indicators of successful intervention without investigating pain behavior [25].

The aim of this study was to evaluate the effects of a calibrated slight exercise protocol on the structural and functional expression of chronic pain in a surgically induced OA model, representing a further step in the validation of this OA model. The hypothesis was that the environmental condition could influence the response to pharmacological treatment of chronic pain, affecting central sensitization and structural joint damage. The specific objectives were: (i) to quantify the analgesic (tactile sensory sensitivity and targeted spinal neuropeptides) and structural (stifle histology) effects induced by forced exercise alone or a multimodal pharmacologic analgesic treatment after OA induction; (ii) to compare the expression of neurotransmitters affected by exercise or multimodal pharmacologic analgesia; and (iii) to test the interaction between exercise and multimodal pharmacologic analgesic treatment. To address these objectives, a prospective, randomized, blinded, and controlled study was conducted in a prevalidated Sprague-Dawley rat model of surgically induced OA [26,27]. The MI-RAT (*M*ontreal *I*nduction of *R*at *A*rthritis *T*esting) model, which combines cranial cruciate ligament transection (CCLT) and destabilization of the medial meniscus (DMM), thus allowing for the progressive development of chronic OA pain based on joint instability and forced mobilization, was compared with sedentary conditions, as previously tested [26].

## 2. Results

### 2.1. Exercise Condition Attenuated Tactile Sensory Sensitivity (Allodynia) in the MI-RAT Model

Analysis of the static quantitative sensory testing (QST) data for the right hind paw (RHP) demonstrated a group effect (*p* < 0.0001), a time effect (*p* < 0.0001), and a time *x* group effect (*p* = 0.03) (Figure 1).

Tactile allodynia (ipsilateral paw) was present immediately at the first timepoint, Day (D) 7 in all groups with lower RHP paw withdrawal threshold (PWT) in sedentary–placebo rats compared with exercise rats (*p* < 0.04) until the end of the experiments (*p* < 0.05). Exercise diminished allodynia development in both placebo and positive analgesic (PA) groups (*p* < 0.04). PA treatment alone in sedentary rats was efficient on D56 only (*p* = 0.001) and had an additive anti-allodynic effect on exercise observed on D21 and D56 (*p* < 0.0001). Combining exercise with PA resulted in a higher anti-allodynic effect compared with sedentary–PA on D21 and D56 (*p* < 0.0001).

On the contralateral hind limb (Figure 2), lower PWT of the left hind paw (LHP) was noted on D7 in sedentary rats compared with exercise rats (*p* < 0.004).

Allodynia continued to develop until the end of the experiment in sedentary–placebo rats compared with exercise rats (*p* < 0.039). No effect of PA treatment was noted in the contralateral hind limb.

Analysis of the QST asymmetry index values revealed a group effect (*p =* 0.049) and a time effect (*p* < 0.0001) but no time *x* group interaction effect (*p* = 0.468) (Figure 3).

On D7, all groups presented marked contralateral sensitivity transfer on the QST asymmetry index, with a mean of –41.73 (3.95)% (*p* > 0.629). This contralateral report tended to disappear in all rats on D56 compared with D7 (*p* < 0.018) but as soon as D21 in the exercise–PA group (*p* = 0.013) compared with the sedentary groups.

### 2.2. Slight Exercise Increases Endogenous Inhibitory Nociceptive Control in the MI-RAT Model

Compared with naïve–ovariectomized (OVX) rats, OA induction in the sedentary–placebo group induced a significant increase in spinal pro-nociceptive tachykinin substance P (SP) and calcitonin gene-related peptide (CGRP) neuropeptides (Table 1).

Exercise decreased pro-nociceptive SP and CGRP in the placebo group compared with the sedentary–placebo group, with a significant difference in SP (*p* = 0.027). Conversely, bradykinin (BK), somatostatin (SST), met-enkephalin (Met-ENK), and leu-enkephalin (Leu-ENK) increased in the exercise–placebo group compared with the sedentary–placebo group (*p* < 0.001), but SST did not reach statistical significance (*p* = 0.180). However, compared with the naïve–OVX group, the exercise–placebo group presented higher pro-nociceptive SP, inflammatory BK, and anti-nociceptive Met-ENK and Leu-ENK spinal concentrations (*p* < 0.002).

Compared with the sedentary–placebo group, the multimodal PA in sedentary OA was associated with a significant decrease in SP (*p* = 0.002) and a significant increase in BK (*p* = 0.004) but no significant changes in other tested neuropeptides. The effect of exercise in the PA group compared with the sedentary–placebo group also translated into significant decreases in pro-nociceptive SP and CGRP (*p* < 0.001) and significant increases in BK, Met-ENK, and Leu-ENK (*p* < 0.048). The addition of PA to exercise compared to exercise–placebo resulted in decreased pro-nociceptive SP and CGRP (*p* < 0.004), as well as decreased anti-nociceptive SST, Met-ENK, and Leu-ENK (*p* < 0.002). Compared with sedentary–PA, SP and CGRP concentrations decreased with exercise, with a significant difference in CGRP (*p* = 0.0002), while Met-ENK increased (*p* = 0.0002) in the exercise–PA group. Finally, the exercise–PA combination was the only condition to get the neuropeptide concentrations to levels similar to those in the naïve–OVX group.

### 2.3. Slight Exercise Exerts a Destructive Influence on Chondral Lesions in the MI-RAT Model

The percentages of cartilage alterations in the total histological modified Mankin’s score (mMs) are provided in Table 2 for comparison.

Histological assessment demonstrated a detrimental effect (*p* < 0.001) of exercise (mMs = 33.3%), with higher cartilage alterations compared with sedentary counterparts (mMs = 12.0%), and with no difference for placebo and PA under both environmental conditions (*p* > 0.378). Exercise increased chondral lesions by +31.1% (4 times increase), proteoglycan loss by +32.3% (3.3 times), and cluster formation by +34.7% (25.8 times); however, chondrocyte loss increased by +12.5% (3.2 times) without exercise (*p* < 0.001).

Experimental conditions affected the repartition (based on 12.0% and 33.3% of total mean cartilage alterations in sedentary and exercise groups, respectively) of different types of cartilage alterations. Indeed, disruption of cellular homeostasis occurred in sedentary rats after the surgical procedure, causing chondrocyte apoptosis/necrosis (34.84 ± 3.95%), chondral lesions (35.26 ± 2.00%), and consequently, alterations in the components of the extracellular matrix (29.15 ± 2.00%), but with low cluster formation (0.75 ± 0.53%) in the tibial alterations of the joints. Chondral lesions (49.49 ± 0.83%) (*p* < 0.001) and cluster formation (13.34 ± 0.83%) (*p* < 0.001) were more predominant under exercise conditions, with no change in proteoglycan loss (33.49 ± 1.07%) (*p* = 0.053) but lower chondrocyte loss contribution (3.69 ± 1.11%) (*p* < 0.001) to the total histological mMs compared with the sedentary condition.

## 3. Discussion

This study investigated the potential influence of environmental conditions (exercise or sedentary) and pharmacological chronic pain treatment (multimodal analgesia or placebo) on pain central sensitization and structural joint damage in a surgical OA rat model. While focusing on targeted spinal neuropeptides, the goal was to characterize the different OA models and determine the one that was most translatable to the human OA condition:Will pain central sensitization be affected by exercise?Will the nociceptive responses to OA pain induction be affected by exercise? In which way?Will the structural damages be affected by slight exercise?How will exercise-induced analgesia compare to pharmacological multimodal analgesia?How will their combination affect responsiveness to treatment?

These questions emerged from our own experience with the chemical monosodium iodoacetate (MIA) (2 mg intra-articular injection) in the OA rat model [26,28,29,30]. Clearly, the functional alterations induced in the MIA OA rat model were fleeting, being maximal D3 post-injection (with a −41.8% average decrease in PWT for *N* = 34 rats) and reverting to baseline values on D21 (only −7.5% difference). Moreover, the structural cartilaginous damages do not evolve from a physiologic (“natural”) process, are focal, dose-dependent, and reach statistical significance compared with a naïve group, from D21 onward [29]. MIA-induced chondrocyte death is aggressive and does not represent the typical progression of OA in humans [25]. For these reasons, we developed and validated the MI-RAT model under sedentary conditions, which confirmed the need to opt for a combination of procedures (CCLT and DMM) inducing stifle instability to generate a persistent functional alteration (tested up to D42) associated with corresponding spinal neuropeptidomics changes [26]. On targeted spinal neuropeptides, compared with naïve–OVX rats, both MIA (on D21) and MI-RAT (on D42) OA models induced a statistically significant increase in SP, CGRP, and SST [26].

Pain-centralized sensitization involves multiple mechanisms, both top-down and bottom-up, all contributing to the hyperresponsiveness of the central nervous system to various inputs. Central sensitization encompasses (over)activation of descending and ascending pain facilitatory pathways [31] and impaired functioning of brain-orchestrated descending antinociceptive (inhibitory) mechanisms [32]. The net result is an increase in nociceptive transmission rather than its inhibition. The results confirmed the hypothesis that a calibrated slight exercise protocol alone would diminish centralized sensory hypersensitivity. This reduction was associated with a decrease in the spinal concentration of pro-nociceptive and an increase in anti-nociceptive neurotransmitters, making it more representative of the human OA pain condition.

### 3.1. Effect of a Calibrated Slight Exercise Training Program on Tactile Sensory Sensitivity, Neuropeptidomics, and Structural Alterations Induced by the Surgical MI-RAT OA Model

Compared with the sedentary–placebo group, the exercise–placebo group presented diminished tactile hypersensitivity (allodynia) as soon as D7, maintained on D21, and brought RHP PWT back toward normality on D56 (Figure 1). Interestingly, the contralateral (right-to-left) report, as assessed using the QST asymmetry index, was similar for all OA groups on D7, identical for both sedentary–placebo and –PA groups, and slightly attenuated by exercise alone on D21 and D56 (Figure 3). This was explained by the absence of tactile sensory sensitivity in the LHP of the exercise–placebo group (Figure 2) at the inverse of the sedentary–placebo group, which manifested LHP tactile hypersensitivity up to D56, suggestive of a more important central sensitization. The latter was counteracted in the sedentary–PA group by the pharmacological multimodal treatment.

The sedentary–placebo MI-RAT group in this study showed, on D56, increased expression of SP, CGRP, and SST identical to that observed in our previous study on D42 post-OA-induction and with half the sample size (*N* = 6). Similarly, the spinal concentration of BK, as well as the concentrations of the endomorphins Met-ENK and Leu-ENK, did not change compared with naïve rats, both previously [26] and in the present study. Such reproducibility in targeted spinal neuroproteomics adds to the validity of the MI-RAT model. At the time of our initial publication on the MI-RAT model [26], we associated the spinal increase in the pro-nociceptive tachykinins SP and CGRP with central neuronal plasticity [28,33,34], and the increased spinal content of SST (379 (45) fmol/mg compared to 227 (39) fmol/mg in the naïve group) to increased descending nociceptive inhibition [35,36]. Calibrated slight exercise diminished the spinal concentration of pro-nociceptive SP and CGRP, possibly through a reinforced descending nociceptive inhibitory control corroborated by the increased spinal concentration of SST and the endomorphins Met-ENK and Leu-ENK [37]. Indeed, SST was previously reported to be the most sensitive OA pain biomarker, with its differential expression associated with pain severity in different surgical [26] and chemical [29] OA rat models. It is logical for SST to follow the evolution of SP and CGRP because it is responsible for inhibiting different cell types capable of producing pro-inflammatory mediators and neuropeptides, including SP and CGRP [38].

Interestingly, exercise led to an increase in BK, and it can be postulated that it is linked to the solicitation of the affected joint’s mobility, thereby stimulating the release of the inflammatory biomarker [39,40,41]. Indeed, the kinin peptide modulates synaptic transmission in the spinal cord, which has a pro-nociceptive role involved in hyperalgesia [42,43] mainly through its polyvalent interactions with other neurotransmitters [44,45,46]. Forced joint mobilization may result in more structural alteration and inflammation [47], thereby supporting both peripheral and central sensitization [48]. These observations support the ability of a slight exercise training protocol to counteract tactile allodynia involving more complex (and natural?) nociceptive modulation and integration, ultimately [24] minimizing contralateral sensitivity transfer in OA rats. The results of environmental conditions (sedentary or exercise) in placebo OA rats on sensory sensitivity phenotype indicated that exercise produced a tonic endogenous analgesia effect. The ability to stimulate the release of endogenous opioid peptides such as β-endorphin and others following aerobic exercise, inducing analgesia while raising the nociceptive threshold, has been well-documented [49,50,51].

Additionally, the pre-emptive administration of opioid antagonist naloxone in normal humans [52,53] and in rats with free access to exercise wheels [54] abolished the exercise-induced increase in the nociceptive threshold. The results of the increased spinal concentration in inhibitory neuropeptides such as enkephalins in the current study support similar mechanisms of pain control. Finally, compared with exercise–placebo, the response to exercise–PA showed that the MI-RAT OA model coupled with a calibrated exercise protocol in rats would be more sensitive and complete enough to detect any analgesic response to tested therapeutic interventions.

Numerous in vivo reports, such as those conducted in the rat chemical OA model, have provided evidence of a direct anti-inflammatory effect of moderate treadmill exercise on stifle OA [21,55,56]. This anti-inflammatory effect seemed associated with the activity of lipoxin A4 and the NF-κB pathway in serum, synovial fluid, and articular cartilage [21]. Moreover, calibrated physical exercise in MIA-induced OA rats resulted in a reduction in oxidative stress and histological preservation of articular cartilage [57]. An exercise training protocol conducted over four weeks also attenuated central pain sensitization development in the chemical OA model by reversing MIA-induced tactile hypersensitivity and body weight asymmetry [58]. The impact of physical therapy exercise on pain assessment has not been extensively documented in surgical OA rat models [25], with the main focus being on histological, structural, and/or molecular changes induced by exercise in these OA models [20,23,47,59,60,61,62,63,64]. While exercise often aggravated structural alterations, exercise training effectively modulated inflammatory processes caused by surgically induced stifle OA, resulting in reduced expression of articular interleukin-1β, caspase-3, and metalloprotease-13 [65]. Slight to moderate exercise had positive structural effects on the severity of chondral lesions in CCLT rats [23,64]. Conversely, a strong effort abolished this chondroprotective effect [23]. Using the DMM model [20,60,61,62], detailed molecular structural benefits of slight exercise intervention were reported.

When looking at the histological assessment, the results first showed that the surgical CCLT–DMM procedure induced some stifle joint instability in rats that translated into chondral lesions. A disruption of cartilage cellular homeostasis occurred in sedentary rats, inducing chondrocyte apoptosis/necrosis, chondral lesions, and consequently, alterations in the components of the extracellular matrix, with loss of proteoglycan. The degeneration and remodeling processes in OA-affected articular cartilage are due to abnormal cell activation [66]. However, poor cluster cell formation was noted, indicating a primitive state of OA progression. Increased number and size of chondrocyte cluster formation are hallmark histological features of OA articular cartilage [66]. Previous studies in rats using a model of surgically induced joint instability have reported similar observations [67,68]. Early events in the post-traumatic OA disease progression showed minimal loss of cartilage proteoglycan and chondrocytes after four weeks [67]. Analysing the repartition of cartilage alteration showed the predominance of chondral lesions and cluster formation after repetitive physical activity, despite lower chondrocyte loss in exercise rats compared with sedentary rats. This was in line with many previous studies, which demonstrated that the inclusion of impact exercise or any additional mechanical load enhanced and accelerated cartilage damage in a different surgically induced model of OA rats, depending on the intensity and duration of the exercise protocol [20,23,59,60,61,62,63,69,70]. Principally, chondrocyte metabolism is based on good diffusion and convection, within the joint, of synovial fluid for an adequate supply of nutrients since the cartilage is an avascular tissue [66]. Abnormal mechanical stress or an overuse of activities can produce cyclic loading and consequently affect the pressure gradients and fluid flow within the tissue, leading to perturbation of chondrocyte metabolism. Our results confirmed that supplemental mechanical stress had a direct impact on chondrocyte metabolism (an anti-apoptotic effect and/or change in chondrocyte behavior) leading more rapidly to a later phase of OA disease progression in exercise groups [23,71]. Finally, it is interesting to note that the multimodal pharmacological treatment did not influence the histological scoring under sedentary or exercise conditions. It must be noted that the structural alterations were assessed on D56, which could have been too early for PA-induced well-being to be translated into increased use of joint and aggravated structural damage.

### 3.2. Synergic Effect of Exercise and Analgesia on Pain in the MI-RAT Model

The combination of exercise and PA resulted in an anti-allodynic effect that could be classified as synergic. On D21, the increase in RHP PWT in the exercise–PA group exceeded the addition of the increase observed in the sedentary–PA and exercise–placebo groups (Figure 1), and the effect was maintained on D56. Notably, only the LHP in the exercise–PA group exhibited hyposensitivity throughout the entire follow-up (Figure 2), and the transfer of contralateral sensitivity was completely reversed as early as D21 (Figure 3). These results suggest that the centralized sensitization was effectively counteracted in the exercise–PA group, eliminating the need to solicit the endogenous inhibitory control system. The neuropeptidomic analysis supports the observations made regarding the functional evaluation of nociceptive sensitization. Additionally, the synergic effect was also observed with SST: the exercise–PA group showed a significant decrease in concentration that the exercise–placebo and sedentary–PA groups were unable to achieve.

### 3.3. Validity of the MI-RAT OA Model Coupled to a Calibrated Slight Exercise Protocol

As presented above, the exercise–placebo group looks promising for studying OA chronic pain in terms of sensory sensitization initiation and persistence, nociceptive modulation and integration processes, and responsiveness to anti-hypersensitive (pregabalin) and anti-inflammatory (carprofen) therapy, as well as more mature joint structural alterations, mimicking better OA human disease. The improved endogenous inhibitory control of pain is hypothesized to facilitate the altered joint manipulation (or the inverse?) with calibrated exercise associated with more substantial structural damage. Another advantage of the MI-RAT OA model coupled with calibrated slight exercise is the fact that the joint lesions are greater and more homogenous (the same standard deviation for an average lesion score was three times higher with exercise vs. sedentary—See Table 2), improving the model’s ability to detect any structural effects of tested disease-modifying OA drugs.

For a pain model to be considered valid, it should encompass key elements of the human pain experience and measure relevant aspects of that experience [72]. The MI-RAT OA model, when coupled with calibrated slight exercise, exhibits face validity by addressing pain sensitization initiation and maintenance, involvement of integrative nociceptive processes, and structural alterations. Furthermore, in a recent review [25], only 10% of the identified literature sources investigated the role of exercise in rat models using female animals, even though almost 60% of people suffering from OA are women [73]. This is a crucial point, as most people impacted by chronic pain are women, and neural pathways involved in developing and maintaining chronic pain are increasingly recognized as being distinct between genders [74]. Evident sexual dimorphism is also reported in the molecular mechanisms of pain sensitization and involves inhibitory control [75]. Unfortunately, this knowledge is still unknown to many researchers. For example, in a recent study of the contribution of CGRP in OA mice stifles, the authors did not mention the sex of the animals used [76]. In addition, our process of validating the MI-RAT model confirmed that sterilized female rats favorably increased data homogeneity by reducing variability in pain outcome measures because estrogen supplementation interfered with central sensitization in OVX rats [27].

The predictive validity was tested with regard to responsiveness to multimodal pharmacological treatment (PA). In sedentary–PA rats, limited changes were observed in neuropeptidomics, with decreased SP (and not CGRP) and increased BK. Could the latter be associated with improved joint manipulation facilitated by the analgesia? The addition of PA to the calibrated slight exercise was translated into further normalization of pro-nociceptive SP, CGRP, and endogenous inhibitory control of SST, Met-ENK, and Leu-ENK. These changes highlight the predictive validity of the model and, compared with the sedentary–placebo group, the increased BK spinal concentration suggests accentuated pain control associated with exercise-induced manipulation of the joint. Exercise-induced antinociception involves various endogenous pain control pathways [12,14,15,16,17] in both healthy and neuropathic rats. As recently suggested [25], there is no doubt that a surgical OA model, such as the MI-RAT, coupled with calibrated slight exercise under standardized procedures (peri-operative analgesia and anesthesia, surgical procedure, enrichment, exercise, outcome measures) in sterilized female rats looks highly promising for clinical translatability of the findings generated.

## 4. Materials and Methods

### 4.1. Animals

Ovariectomized (OVX) Sprague-Dawley rats (*N* = 60) (230–250 g, 6–8 w), provided by Charles River Laboratories, Canada (Saint-Constant, QC, Canada), were housed under regular laboratory conditions with a constant temperature of 22 °C in a 12 h light–dark cycle. Food and water were provided ad libitum. The study was conducted in an enriched environment and care, which included two rats per cage, cardboard boxes, pipes, and fruit crunchy treats, was provided according to the facility’s standardized operating procedure (SOP AC7011-3).

The care and use of animals were subject to and approved by the “Comité d’Éthique de l’Utilisation des Animaux de l’Université de Montréal” (N° Rech-1766). The study was conducted per the principles outlined in the current Guide to the Care and Use of Experimental Animals published by the Canadian Council on Animal Care and the Guide for the Care and Use of Laboratory Animals published by the US National Institutes of Health, complying with the ARRIVE guidelines.

### 4.2. Experimental Protocol

A naïve group (*N* = 12) of OVX rats was used as a control to compare the baseline values of all rats, with repeated measurement (single for neuropeptidomics) of this group, to those measured in OA-induced groups.

#### 4.2.1. Induction of OA

After surgical preparation on D0, right stifle instability was performed in the rats following a CCLT–DMM procedure, as previously described [26,27]. Before surgery (40 min before general anesthesia was administered), an intramuscular injection of buprenorphine (Buprenorphine SR^®^, Chiron Compounding Pharmacy Inc., Guelph, ON, Canada) was administered at a 1.0 mg/kg dose. Anesthesia was induced using 2% isoflurane (IsoFlo^®^, Abbott Animal Health, Saint-Laurent, QC, Canada) in an O_2_ mixture. At the end of the surgical procedure, a 0.25% peri-incisional block of bupivacaine (Marcaine^®^, McKesson Canada, St.-Laurent, QC, Canada) was administrated at a dose of 1.0 mg/kg (0.05–0.1 mL per site).

#### 4.2.2. Analgesic Treatment and Exercise Protocols

After OA induction on D0, 48 female OVX rats were randomly assigned to four equal groups (*N* = 12) under sedentary or exercise conditions under PA or negative (placebo) control. From D3 to D56, rats in both PA treatment groups received pregabalin (30 mg/kg) combined with carprofen (5 mg/kg) via daily subcutaneous injections [30,77]. Negative control groups included rats receiving a daily subcutaneous injection of vehicle placebo (at an equivalent of 0.9% sterile saline volume of 1 mL/kg). Exercise conditions included a 5 min session of forced motor activity (at 12 revolutions per minute) on a Rotarod (EzRod, Omnitech Electronics, Inc, Columbus, OH, USA) followed by 10 min of running on a motor-driven treadmill for rodents (IITC Life Science Inc., Woodland Hills, CA, USA) at a constant speed (18.3 cm/s). No stimulus was required to promote movement, and positive reinforcement was systematically used, with treats and vegetables during exercise (treadmill and forced motor activity) and a daily interactive play period between the manipulator and the rat as recommended [78]. A total distance of 2.54 km was covered by each animal in the exercise groups during the entire experiment (total of 21 days). The exercise was considered to be slight when the total distance covered was less than 7.5 km over 28 days [23]. An acceptable cutoff was under 2.0 km/day [78]. After 2 weeks of gradual and progressive acclimatization, the exercise protocol was conducted on three non-consecutive days a week for eight weeks from D3 to D56.

### 4.3. Functional Assessment of Nociception

Testing was performed during the daylight phase and was undertaken by the same two female observers who were blinded to treatment group allocation. All animals were allowed to acclimate to testing conditions (QST and exercise protocol) according to a validated acclimation protocol based on five days of exposure to the apparatus over two weeks before induction of OA, as previously published [27,28,29,30]. Functional assessments were performed one day before the induction of OA for baseline values (D–1) and on D7, D21, and D56 post-induction of OA pain.

#### Static QST

Rats were placed inside an elevated metal grid cage and allowed an exploration session of 2 min. Secondary tactile sensory sensitivity was assessed using an Electronic von Frey Esthesiometer^®^ (IITC Life Science Inc., Woodland Hills, CA, USA) with a standardized filament (0.7mm^2^ polypropylene Supertip) to obtain punctate tactile PWT expressed in grams. Both hind paws were tested three times in a random order, with a refractory period of one minute between each trial [26,27,28,29]. The QST asymmetry index between the ipsilateral and contralateral hind limbs was calculated in percentage (%) as follows:(|PWT_right_ − PWT_left_|/[|PWT_right_ + PWT_left_| × 0.5]) × 100

The static QST detects tactile sensory sensitivity development associated with secondary allodynia and hyperalgesia, recognized as a clinical expression of nociceptive sensitization [79].

### 4.4. Neuropeptidomic Analysis

#### 4.4.1. Spinal Cord Sample Preparation

Euthanasia was performed by decapitation following isoflurane overdose after the last functional evaluation day (D56), after which collection of the whole spinal cord was achieved using a saline flush technique [26,27,28,29,30]. Samples were snap-frozen in cold hexane, stored individually, and kept at −80 °C pending neuropeptidomic analysis. All chemicals were obtained from Sigma-Aldrich (Oakville, ON, Canada), except where expressly indicated.

Rat spinal cords were individually weighed and homogenized, as previously published [28]. The samples were sonicated for 20 min and high molecular weight proteins were removed. The homogenate was then mixed with acetonitrile in a 1:1 (*v/v*) ratio. The samples were vortexed and centrifuged for 10 min at 12,000 × *g*. The supernatant was then transferred into a microcentrifuge tube and spiked with an internal standard solution containing 50 pmol/mL of labeled targeted neuropeptides in 0.1% 2,2,2-trifluoroacetic acid (TFA) in water in a ratio of 1:1 (*v/v*).

Peptides were extracted using a standard C18 solid phase extraction protocol. Agilent Bond Elut C18 (100 mg/1 mL) was activated with 1 mL of 50% acetonitrile in water. The cartridge was equilibrated twice with 1 mL of a 5% acetonitrile solution in water containing 0.2% TFA. Using positive displacement, 400 µL of sample was loaded slowly onto the cartridge. The resin was washed with 1 mL of a 5% acetonitrile solution in water containing 0.2% TFA. Peptides were eluted twice using 750 µL of 65% acetonitrile solution in water containing 0.2% TFA. The resulting residues were dried using an Eppendorf vacuum concentrator, resuspended in 50 µL of 5% acetonitrile in water containing 0.1% formic acid, and transferred into low-volume, high-performance liquid chromatography (HPLC) vials.

#### 4.4.2. Chromatographic Conditions

Chromatography was performed using a gradient mobile phase along with a microbore column Thermo Biobasic C18 100 × 1 mm with a particle size of 5 μm using an UltiMate 3000 Rapid Separation UHPLC system (Thermo Scientific, San Jose, CA, USA). The initial mobile phase condition consisted of acetonitrile and water (both fortified with 0.1% of formic acid) in a ratio of 5:95. From 0 to 1 min, the ratio was maintained at 5:95. From 1 to 21 min, a linear gradient was applied up to a ratio of 50:50, which was maintained for 4 minutes. The mobile phase composition ratio was reverted to that at the initial conditions and the column was allowed to re-equilibrate for 15 min for a total run time of 40 min. The flow rate was fixed at 75 µL/min and 2 µL of sample was injected.

#### 4.4.3. Mass Spectrometry Conditions

A Q Exactive Plus Orbitrap Mass Spectrometer (Thermo Scientific, San Jose, CA, USA) was interfaced with an UltiMate 3000 Rapid Separation UHPLC system using a pneumatic-assisted heated electrospray ion source. Mass spectrometry detection was performed in positive ion mode and operated in full scan mode at high resolution and accurate mass. Nitrogen was used as the sheath and auxiliary gas and was set to 10 and 5 arbitrary units. The heated ESI probe was set to 4000 V and the ion transfer tube temperature was set to 300 °C. The scan range was set to *m/z* 400–1500. Data were acquired at a resolving power of 140,000 (FWHM) using an automatic gain control target of 3.0 × 10^6^ and a maximum ion injection time of 200 msec. Targeted peptide quantification was performed at the MS^1^ level using specific precursor masses based on the monoisotopic masses. Peptide quantification was performed by extracting specific precursor ions using a 5ppm mass window. Instrument calibration was performed prior to all analyses, and mass accuracy was notably below 1 ppm based on the Thermo Pierce calibration solution and automated instrument protocol. SP, CGRP, BK, SST, Met-ENK, and Leu-ENK peptide quantification was performed using a peak-area ratio of light (unlabeled) and heavy (labeled) isotopes and expressed in fmol/mg of spinal cord homogenate, as previously described [26,27,28,29,30,67].

### 4.5. Histological Analysis

Right stifle joints were collected and fixed in a 10% formaldehyde solution (pH 7.4) for at least three days after being dissected free of muscle. Following fixation, stifle joints were decalcified and embedded in paraffin. For each stifle, serial sections with a thickness of 5 µm were cut for histological examination after a Safranin O–fast green staining. The severity of OA articular lesions, ranging from 0 to 25, was graded on a scale using the modified Mankin’s score (mMs) table [33,80,81] (see Table A1) by two independent observers (a certified veterinary pathologist and an experienced pathology technician) blinded to the study. Structural changes to assess chondral lesions were graded from 0 (normal) to 10 (highest surface). Safranin O staining was scored from 0 (no loss of staining) to 6 (more than 50% loss of staining in all the articular cartilages) to identify proteoglycan loss. The formation of clusters was evaluated using a scale from 0 (no cluster formation) to 3 (more than 8 clusters). Finally, the cellularity of chondrocytes was assessed for loss of chondrocytes from 0 (normal) to 6 (diffuse). This score was determined using two compartments of the ipsilateral stifle, the medial and lateral parts of the tibia, thus resulting in a maximal score of 50 per stifle (representing 100% of alterations), expressed in percentage of cartilage alterations.

### 4.6. Statistical Analysis

All statistical analyses were conducted using SPSS statistical software (IBM^®^ SPSS^®^ Statistics Server version 23.0, New York, NY, USA). Briefly, the average of the three trials for QST data (PWT and QST asymmetry index) was analyzed using general linear mixed models for repeated measures and the normality of the outcomes was verified using the Shapiro–Wilk test. Treatment groups, day, and their interactions (day *x* group) were considered fixed effects with baseline measurements as covariates and were analyzed using the type 1 regressive covariance structure. Neuropeptide quantification and histological mMs were analyzed using the non-parametric Mann–Whitney–Wilcoxon test to identify differences between groups following a non-parametric Kruskal–Wallis one-way analysis of variance. Data are presented as the estimated mean (least square mean (LSM)) with 95% confidence limits (inferior and superior) for QST data or mean (standard deviation) for histological mMs analysis, as well as median with minimum and maximum for the concentration of neuropeptides measures. The threshold alpha for inferential analysis was set at 5%.

## 5. Conclusions

These findings highlighted the benefits of a calibrated slight exercise protocol in an MI-RAT surgical OA model, improving its validity for pain assessment, structural alterations, and clinical translatability. They also confirmed the analgesic effect of exercise on chronic OA pain induced in a preclinical model. The exercise-associated reduction in sensory hypersensitivity was associated with increased endogenous inhibitory nociceptive control and attenuation of pain facilitatory pathways, allowing for a more complete pain profile to be studied. The PA protocol was not associated with increased structural alterations under sedentary conditions, an effect usually associated with joint supplemental solicitation with PA-related comfort. The anti-allodynic property of exercise was associated with improved comfort and additional use of the affected limb. The synergism between exercise and PA in stopping pain development was evident, supporting the responsiveness of the MI-RAT OA model coupled with a calibrated slight exercise to analgesics.

## Figures and Tables

**Figure 1 ijms-24-16341-f001:**
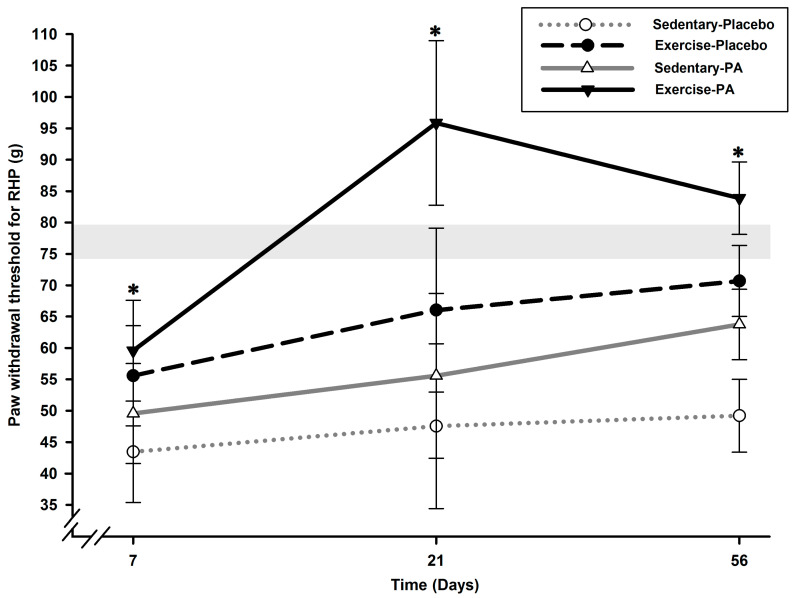
Static QST of ipsilateral RHP PWT after surgically induced OA pain in rats (*N* = 12/group). Allodynia (ipsilateral paw) was present as soon as D7 in all groups and was maintained up to D56 in sedentary–placebo rats with a lower PWT compared with exercise rats (*p* < 0.05). Exercise counteracted it in both placebo and PA groups (*p* < 0.04). PA treatment alone was efficient on D56 only (*p* = 0.001) but presented an additive anti-allodynic effect to exercise (D21; D56; *p* < 0.0001). * Significant statistical inter-group differences for each time point. Least squares mean ± 95% confidence limit intervals are represented. The horizontal gray zone represents baseline values (D–1) and naïve rats (mean (standard deviation)) for RHP PWT static QST (76.85 (2.42) g).

**Figure 2 ijms-24-16341-f002:**
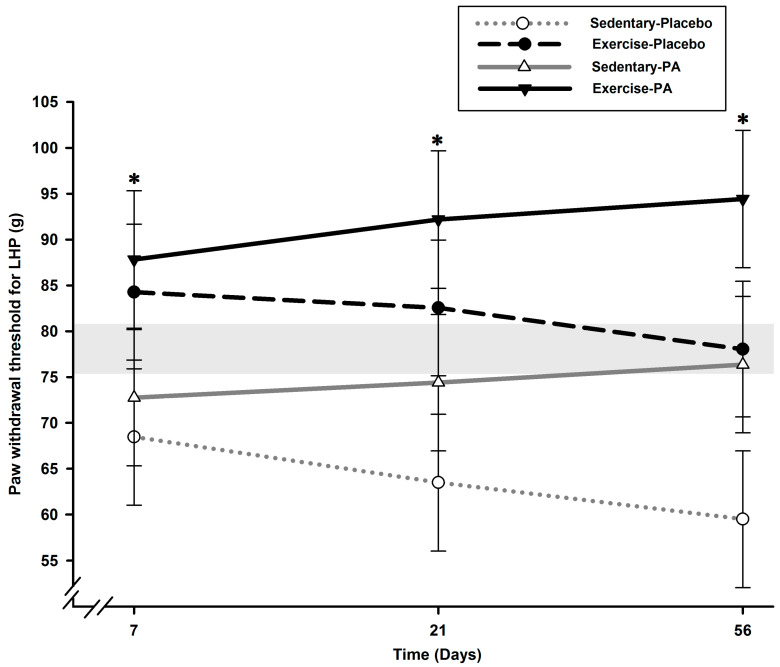
Static QST of contralateral LHP PWT over time following surgically induced osteoarthritis pain in rats (*N* = 12/group). Lower PWT of LHP was noted on D7 in sedentary rats compared with exercise rats (*p* < 0.004). Allodynia continued to develop in sedentary–placebo rats until the end of the experiment (*p* < 0.039). No effect of PA treatment was noted in the contralateral hind limb. * Significant statistical inter-group differences for each time point. Least squares mean ± 95% confidence limit intervals are represented. The horizontal gray zone represents baseline values (D–1) and naïve rats (mean (standard deviation)) for LHP PWT static QST (78.61 (2.90) g).

**Figure 3 ijms-24-16341-f003:**
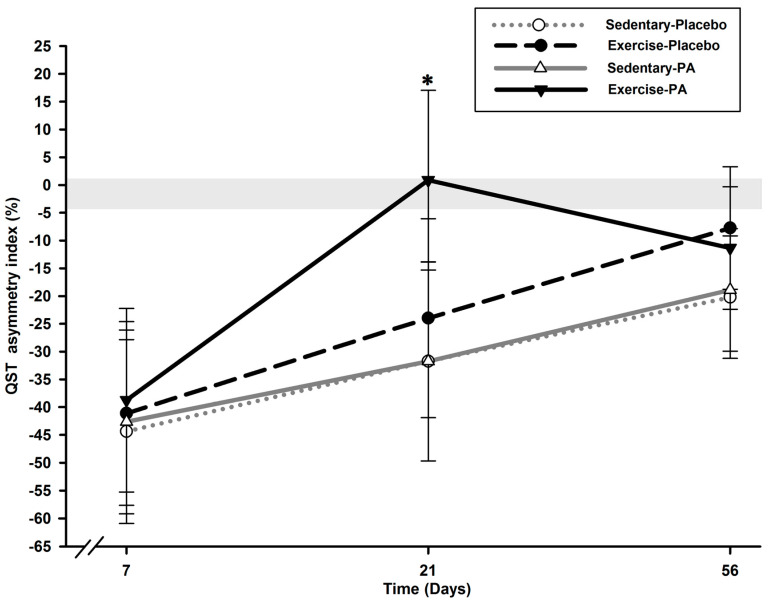
Temporal evolution of static QST asymmetry index between ipsilateral and contralateral hind limbs over time. On the QST asymmetry index, all groups (*N* = 12 rats/group) presented marked contralateral transfer on D7, with a mean of −41.73 (3.95)%, which disappeared in all groups on D56 (*p* < 0.018) but as soon as D21 in the exercise–PA group (*p* = 0.013). * Significant statistical inter-group differences for each time point. Least squares mean ± 95% confidence limit intervals are represented. The horizontal gray zone represents the QST asymmetry index at baseline (D–1) and naïve rats (–1.45 (3.11)%).

**Table 1 ijms-24-16341-t001:** Spinal concentration of neuropeptides measured on day 56 of follow-up.

Experimental Groups ^a^	Neuropeptides (fmol/mg)					
	SP	CGRP	BK	SST	Met-ENK	Leu-ENK
Naïve–OVX	79.9	451.7	237.9	324.6	73.2	55.2
	(75.0; 83.0)	(370.3; 532.4)	(187.3; 262.3)	(295.5; 466.1)	(62.3; 78.1)	(51.5; 61.7)
Sedentary–Placebo	123.9 *	596.1 *	199.1	352.5	68.0	46.8
	(98.5; 209.0)	(464.5; 679.0)	(162.7; 240.5)	(286.2; 439.2)	(45.1; 90.7)	(30.4; 63.2)
Exercise–Placebo	102.4 *^,^**	498.0	312.78 *^,^**	419.9	97.5 *^,^**	74.9 *^,^**
	(91.9; 115.3)	(447.4; 664.8)	(273.0; 354.3)	(345.9; 429.4)	(93.1; 108.8)	(68.6; 79.9)
Sedentary–PA	88.6 ***	576.4 *	254.8 ***	347.0	62.9 *	62.2
	(58.5; 119.6)	(488.2; 620.0)	(195.0; 349.0)	(286.4; 489.0)	(42.3; 70.1)	(40.0; 81.4)
Exercise–PA	77.2 ^¥,§^	399.9 ^¥,§,Ŧ^	263.0 ^¥^	334.1 ^§^	82.3 ^¥,§,Ŧ^	59.6 ^¥,§^
	(54.7; 81.3)	(364.6; 469.9)	(204.0; 322.4)	(305.6; 344.4)	(72.8; 90.6)	(48.2; 67.4)

^a^ After surgical OA induction, all rats (*N* = 12/group), except the naïve–OVX group, received daily single doses of positive analgesia (exercise–PA and sedentary–PA groups) or placebo (exercise–placebo and sedentary–placebo groups) from D3 to D56 and were, or not, exposed to a training protocol (exercise or sedentary conditions, respectively). * Significant between-group differences (any of the four MI-RAT groups) compared with the naïve–OVX group. ** Exercise–placebo group was statistically different from the sedentary–placebo group. *** Sedentary–PA group was statistically different from the sedentary–placebo group. ^¥^ Exercise–PA group was statistically different from the sedentary–placebo group. ^§^ Exercise–PA group was statistically different from the exercise–placebo group. ^Ŧ^ Exercise–PA group was statistically different from the sedentary–PA group.

**Table 2 ijms-24-16341-t002:** Histological modified Mankin’s scores for percentage of cartilage alterations in the tibia right hind stifle.

^a^ Experimental Groups	^b^ mMs (%)	
	Mean (%)	SD (%)
**Total cartilage alterations (%)**		
Exercise–Placebo	35.7	10.9
Exercise–PA	31.0	6.0
**Mean Exercise groups**	33.3	4.4
Sedentary–Placebo	11.2	7.5
Sedentary–PA	13.0	9.7
**Mean Sedentary groups**	12.0	4.2
**Cartilage alterations (%) for each criteria**		
**Exercise groups**		
Chondral lesions	41.5	12.8
Proteoglycan loss	46.2	12.8
Cluster formation	36.1	10.6
Chondrocyte loss	5.6	7.6
**Sedentary groups**		
Chondral lesions	10.4	8.1
Proteoglycan loss	13.9	9.7
Cluster formation	1.4	4.7
Chondrocyte loss	18.1	13.6

^a^ All rats (*N* = 12/group) received a single daily dose of positive analgesia (exercise–PA and sedentary–PA groups) or placebo (exercise–placebo and sedentary–placebo groups) from D3 to D56 following surgically induced OA pain on D0 and were, or not, subjected to a training protocol (exercise or sedentary conditions, respectively). ^b^ This measure was obtained for the four groups in the percentage of cartilage alterations for the histological mMs of the medial and lateral right tibial compartments at sacrifice.

## Data Availability

The data presented in this study are available on request from the corresponding author.

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
