# Peer review of "Face and Predictive Validity of MI-RAT (Montreal Induction of Rat Arthritis Testing), a Surgical Model of Osteoarthritis Pain in Rodents Combined with Calibrated Exercise"

_ijms, 2023, doi:10.3390/ijms242216341_

Round 1

Reviewer 1 Report

Comments and Suggestions for Authors

C. O. et al. present a new experimental model, the MI-RAT, for the assessment of osteoarthritic pain. The topic is of scientific interest because a bidirectional relationship may exist between joint damage and pain. Over the past two decades, several combined electrophysiological and neurobiological studies have been conducted to better understand peripheral and central sensitisation to mechanical stimuli.

The authors used ovariectomised female Sprague-Dawley rats to study the effects of analgesic treatment (pregabalin+carprofen) and calibrated physical exercise on nociception. They surgically induced OA and recorded the development of allodynia, neuropeptide concentrations isolated from the spinal cord and histological signs on days 7, 21 and 56. The main findings of the study are:

- Allodynia appeared from day 7 and was maintained until day 56 in sedentary-placebo rats and was counteracted by exercise in animals with and without analgesic treatment,

- These effects were not observed in the contralateral limb,

- The static quantitative sensory test asymmetry index showed a contralateral shift on day 7, which disappeared on days 21 and 56 in the exercise-analgesic group,

- Three pro-nociceptive SP and CGRP peptides decreased in the exercise-placebo and exercise-PA groups, whereas Met-Enk showed an increase,

- Exercise had a detrimental effect on histological parameters, except for chondrocyte loss, which was inhibited compared to sedentary animals.

This is a carefully designed study with some interesting results. However, before validating the results and publishing the manuscript, some improvements should be made and some questions should be answered:

1. The level of exercise calibrated should be justified. The exercise protocol was performed according to Galois et al.[1], and in general follows the guidelines of Poole et al.[2]. However, it should be explained why it is considered to be only a light exercise, and it should be compared energetically to average physical efforts of the rat (in W, or cal).

2. The histological results are interesting, but need more explanation. The authors must cite, which source did they use for the modified Mankin’s score. They should explain, how each parameter was exactly defined. Moreover, they have to justify, why did they not apply the better standardized and recommended OARSI histological guidelines for evaluation [3]. For a better visual control, they should rearrange Table 2., pairing the results obtained at sedentary and exercise animals.

3. Table 1. is also difficult to understand in terms of comparisons for statistical significance. The authors should introduce an explanatory footnote, what groups the a,b,c,d comparisons refer to.

Otherwise, the results are valuable, the discussions are comprehensive and the study has sufficient merits for publication.

References

1. Galois, L.; Etienne, S.; Grossin, L.; Watrin-Pinzano, A.; Cournil-Henrionnet, C.; Loeuille, D.; Netter, P.; Mainard, D.; Gillet, P. 636 Dose-response relationship for exercise on severity of experimental osteoarthritis in rats: a pilot study. Osteoarthritis Cartilage 637 2004, 12, 779–786.

2. Poole DC, Copp SW, Colburn TD, Craig JC, Allen DL, Sturek M, O'Leary DS, Zucker IH, Musch TI. Guidelines for animal exercise and training protocols for cardiovascular studies. Am J Physiol Heart Circ Physiol. 2020 May 1;318(5):H1100-H1138. doi: 10.1152/ajpheart.00697.2019. 

3. Gerwin N, Bendele AM, Glasson S, Carlson CS. The OARSI histopathology initiative - recommendations for histological assessments of osteoarthritis in the rat. Osteoarthritis Cartilage. 2010 Oct;18 Suppl 3:S24-34. doi: 10.1016/j.joca.2010.05.030

Author Response

Reviewer #1

General comment:

C. O. et al. present a new experimental model, the MI-RAT, for the assessment of osteoarthritic pain. The topic is of scientific interest because a bidirectional relationship may exist between joint damage and pain. Over the past two decades, several combined electrophysiological and neurobiological studies have been conducted to better understand peripheral and central sensitisation to mechanical stimuli.

The authors used ovariectomised female Sprague-Dawley rats to study the effects of analgesic treatment (pregabalin+carprofen) and calibrated physical exercise on nociception. They surgically induced OA and recorded the development of allodynia, neuropeptide concentrations isolated from the spinal cord and histological signs on days 7, 21 and 56. The main findings of the study are:

- Allodynia appeared from day 7 and was maintained until day 56 in sedentary-placebo rats and was counteracted by exercise in animals with and without analgesic treatment,

- These effects were not observed in the contralateral limb,

- The static quantitative sensory test asymmetry index showed a contralateral shift on day 7, which disappeared on days 21 and 56 in the exercise-analgesic group,

- Three pro-nociceptive SP and CGRP peptides decreased in the exercise-placebo and exercise-PA groups, whereas Met-Enk showed an increase,

- Exercise had a detrimental effect on histological parameters, except for chondrocyte loss, which was inhibited compared to sedentary animals.

This is a carefully designed study with some interesting results. However, before validating the results and publishing the manuscript, some improvements should be made and some questions should be answered.

Response to general comment:

The authors greatly acknowledge the Reviewer #1 for his/her exhaustive analysis of the present manuscript and the general comments that effectively summarizes our study’s objectives and results. However, we would like to clarify the objective that the study was NOT designed to test the effect of an exercise protocol on the surgical model. The MI-RAT model includes surgical induction of joint (cartilage, mostly) damages and their homogenization in expression with the imposed exercise protocol.  Detailed responses to specific inquiries can be found below.

Comment #1:

            The level of exercise calibrated should be justified. The exercise protocol was performed according to Galois et al., 2004, and in general follows the guidelines of Poole et al., 2020. However, it should be explained why it is considered to be only a light exercise, and it should be compared energetically to average physical efforts of the rat (in W, or cal).

Response no.1:

We thank the reviewer for their comment. Although, we extensively discussed the various impacts of different degree of exercise protocol previously demonstrated by other research teams in the introduction (lines 76 to 83), it appears that the justification for our choice of a slight exercise protocol may not be entirely clear.

Our goal was to develop an animal model that closely mimics human pathology of naturally occurring OA chronic pain which includes slow and progressive cartilage degeneration and not too much aggressive. It has been demonstrated that a calibrated slight or moderate exercise could have a positive influence on the severity of chondral lesions in an experimental surgically-induced OA model [1,2]. Conversely, a strong effort could abolish this effect suggesting a dose-response relationship for exercise on structural lesion severity in experimental models [3] or for naturally-occurring OA in naïve rats [4,5]. Also, as mentioned in discussion part (lines 385-389), in our study, slight exercise protocol has increased the homogeneity of lesions, resulting in higher translational value of our results and therefore, increasing the validity of MI-RAT OA model.

Because Galois et al., 2004 [1] demonstrated a dose-response relationship of exercise in terms of distance covered during a specified time-period in a surgically induced OA model, it was entirely justified and relevant to apply the same classification (< 7.5 km in 28 days = slight exercise), because the different impacts have already been established and demonstrated before for laboratory rats in the same range of age (6-8 w) and body weight (200-250 g). This looked to us more intuitive to adopt this distance classification for a musculoskeletal disease audience than work (W) or calories. For example, the general rule is that a body will burn 1 calorie per kg (BW), per km. Therefore, a 240 g rat running 2.5 km will burn 10 cal. We are not sure this information to be attractive for our readers, when a distance is measured with precision. In Poole et al., 2020 [6], there is no recommendation to report and compare it to the average physical efforts of the rat. As mentioned by Reviewer #1, many recommendations cited in that guidelines have been applied in our animal study. We are grateful that he/she noticed them.

Nevertheless, we have modified the section 4.2.2 (lines 464-471) of Material and Methods part to include additional details to avoid any confusion and enhance clarity in the updated manuscript.

Comment #2:

             The histological results are interesting but need more explanation. The authors must cite, which source did they use for the modified Mankin’s score. They should explain, how each parameter was exactly defined. Moreover, they have to justify, why did they not apply the better standardized and recommended OARSI histological guidelines for evaluation (Gerwin et al., 2010). For a better visual control, they should rearrange Table 2., pairing the results obtained at sedentary and exercise animals.

Response no.2:

In response to this remark, we have included the complete and detailed modified Mankin’s scoring table (adapted from [7-9]) used to assess the cartilage alteration of the stifle in Appendix A (Table A1). Additionally, we have provided the corresponding references in section 4.2 (line 557) of the manuscript. The table A1 already incorporates OARSI histological guidelines published by Gerwin et al., 2010 [9]. Furthermore, one of the blinded observers who scored the stifle was an experienced veterinary pathologist (A.B.) with extensive expertise in histopathological evaluation and a strong familiarity with the use of the modified Mankin’s scoring table. The latter is recognised as focusing specifically on cartilaginous damages when the OARSI scale is used for whole joint alterations. In this validation step of our model, we did hypothesize the cartilage to be the most demonstrative section for induced lesions. Finally, Table 2 in section 2.3 have been enhanced to improve the visual presentation of the histological results.

Comment #3:

            Table 1. is also difficult to understand in terms of comparisons for statistical significance. The authors should introduce an explanatory footnote, what groups the a,b,c,d comparisons refer to.

Response no.3:

We appreciate the Reviewer's valuable remark and apologize for any inconvenience it may have caused. In the revised manuscript, we have reviewed and modified the annotation in Table 1 for statistical significance. This includes the addition of footnotes with specific explanations (lines 168-172) for each between-groups comparisons.

Comment #4:

            Otherwise, the results are valuable, the discussions are comprehensive, and the study has sufficient merits for publication.

Response no.4:

The authors are deeply grateful for the reviewer's positive feedback regarding the relevance of publishing these results. We share his/her enthusiasm for making our work accessible to other members of the scientific community, with the aim of fostering further advancements in translational research in osteoarthritis, and we remain dedicated to delivering high-quality research.

Reviewer 2 Report

Comments and Suggestions for Authors

This is an interesting study describing pain and histological outcomes of a combined exercise and multimodal analgesic treatment of a novel surgical OA model using sterile female rats. The results suggest improved pain outcomes despite worsened histological damage using combined treatment. This effect may be mediated by differential regulation of neuropeptides. Authors suggest that the novel model is particularly valuable for translational pain research - in particular chronic pain in female OA patients. The results presented here are of interest to the broader OA research community.

However, I have several concerns that need to be addressed:

1) The title does not seem to reflect the aim of the study. The model has been established (ref 26) and used (ref 27) previously. The study design in ref 27 is very similar to this study, yet omitting estrogen supplementation. It does not seem appropriate to investigate face and predictive validity here, when the model has already been used extensively. It is unclear whether authors seek to promote the use of a novel animal model (MI-RAT) or to investigate effects of combined excercise and analgesic treatment in this model. I think the latter is more interesting for the OA research community.

2) Sample sizes for the allodynia assessment, neuropeptide quantification and histological assessments are not indicated. I assume these were n=12 per group, but it is conceivable that sample sizes were smaller for the molecular and histological analyses. Please mention sample sizes in the figure legends.

3) The symbols indicating between-group differences in Table 1 are very difficult to interpret. I assume that a,b,c,d refers to the experimental groups 2-4 in the Table? Please indicate in the legend what between-group difference is meant here and check if these are correct. 

4) I am not sure the statement in line 178-180 is correct. None of the naive-OVX values seem to differ significantly from sedentary-PA (letter c) or exercise-PA (letter d). I understand it is tempting to explain the improved pain outcome by neuropeptide levels returning to baseline, but this also happens in sedentary-PA groups. Please rephrase and discuss whether other mechanisms (muscular?) may have contributed to improved pain outcomes.

5) It is unconventional to express modified Mankin score as percentage of cartilage alterations (Table 2 - line 514 - 528) and I have difficulties interpreting these values. I assume the maximal score of 50 (medial and lateral tibia) would represent 100% cartilage alteration? Authors have not followed the OARSI recommendations (ref 80) as claimed in line 520. I am not aware of validated scoring systems that have 10 subgrades for structural changes, 6 for proteoglycan loss and 6 for chondrocyte cellularity. Please use validated scoring systems to assess histological changes such as PMID: 11924812. Authors should keep in mind that histological scores are to be treated as non-parametric data and using percentages is not meaningful. Between-group differences should be evaluated using non-parametric testing on scores, rather than percentages.

6) The discussion should be shortened where possible. Authors should refrain from summarizing results (e.g. line 343-355). I suggest focussing the discussion on the following aspects i) What are the advantages of the novel model compared to other experimental OA rat models? ii) What factors other than neuropeptides may have contributed to better pain outcomes by combined treatment and iii) What is the potential translational value of the model, a specific OA population perhaps?

I hope my comments and suggestions are deemed useful.   

Comments on the Quality of English Language

1) Several sentences are overly wordy (e.g. lines 53, 55, 55-60, 63-66, 172-176, 197-202). The overall readability of the manuscript could be improved by shortening sentences. AI-writing tools such as ChatGPT could be helpful to correct both grammar and readability using the prompt "Check grammar and improve readability". 

2) An AI-based grammar check may also help with incorrect translations from French into English.   

Author Response

Reviewer #2

General comment:

            This is an interesting study describing pain and histological outcomes of a combined exercise and multimodal analgesic treatment of a novel surgical OA model using sterile female rats. The results suggest improved pain outcomes despite worsened histological damage using combined treatment. This effect may be mediated by differential regulation of neuropeptides. Authors suggest that the novel model is particularly valuable for translational pain research - in particular chronic pain in female OA patients. The results presented here are of interest to the broader OA research community.

Response to general comment:

                   The authors are thankful for the Reviewer's #2 appreciation of the current manuscript, and we will make the necessary adjustments to address the following issues to meet their expectations. Just a correction that we would not qualify these female rats as “sterile” but as “sterilized”. This was a surgical sterilization, a veterinary act. Moreover, we would not qualify the observed changes in pain outcomes as an “improvement” but more as a “better pain control” more translating to the reality of chronic pain in osteoarthritic patients.

Comment #1:

The title does not seem to reflect the aim of the study. The model has been established (ref 26) and used (ref 27) previously. The study design in ref 27 is very similar to this study, yet omitting estrogen supplementation. It does not seem appropriate to investigate face and predictive validity here, when the model has already been used extensively. It is unclear whether authors seek to promote the use of a novel animal model (MI-RAT) or to investigate effects of combined exercise and analgesic treatment in this model. I think the latter is more interesting for the OA research community.

Response no.1:

The authors thank the reviewer for his/her comment. It has come to our attention that certain points may not be completely clear, and we sincerely apologize for any lack of precision. Therefore, we believe it is important and crucial to resume and further explain the purpose of the two previous studies.

In Gervais et al., 2019 [10], it was the first demonstration, involving separate or combined, cranial cruciate ligament transection and destabilization of the medial meniscus, of their ability as surgical models to allow a progressive development of OA over time. The combination was the unique model validated as such. The latter was also associated with persistent chronic pain changes similar to the human pathology. It's also important to mention that this pilot study included a very small sample size, only N=6 animals per group, which contributed to a low statistical power and an increased risk of type I statistical error. This was indeed the development of the MI-RAT surgical model and the first step in validation of the model. It was, therefore, important to be able to reproduce and repeat the model while also incorporating the assessment of other factors that could affect pain induction, expression and … evaluation.

In the second study, Keita-Alassane et al., 2022 [11], the effect of 17β-estradiol supplementation in female neutral rats has been evaluated in MI-RAT model since gender dimorphism must also be carefully considered when evaluating pain in rodent. Several studies have explored the sex differences in basal nociceptive sensitivity in rodents [12]. The conflicting results reported can potentially be attributed to the genotype of the experimental animals, variations in experimental settings (such as outcomes and assays), or limitations in statistical power [13,14]. This study confirmed only our hypothesis of a potential sexual hormones interference, particularly oestrogens, and sustained our conclusion to use in the future only ovariectomized (OVX) female rats for assuring more reproducible and homogenized results in OA pain assessment. This was the second step in validation of the model.

The lack of validity in animal models and/or in chronic pain assessment methods participated in the incapacity to translate promising intervention from animal models to clinical applications [15]. Validation is an ever-on-going process, and the reproducibility in results (present in this study) is a fundamental part of it. Then, clearly, by exposing the MI-RAT model to a calibrated exercise protocol, we aimed to emphasize the importance of ongoing refinement when using animal models to increase the predictability translational aspect to clinical applications. This, in turn, contributes to improving the face and predictive validity of the MI-RAT model (including OVX, surgical induction and calibrated exercise protocol). In this case, the title accurately reflects the core concept of the manuscript.

This is the third step in our validation process. The previous steps focused on the development of the model and its concurrent validation to another existing model, the most popular being MIA chemical model. We appreciate the Reviewer’s proposition about presenting the manuscript as reflecting the synergic association between slight calibrated exercise protocol and multimodal PA to fight the chronic pain generated by our MI-RAT OA model. However, in our mind, to reach such goal, we would need to test different degrees of exercise, and more than only one PA protocol. For this reason, we are not comfortable in such presentation, and we would prefer to keep our initial goal of face and predictive validity of our complete (surgical induction in OVX female rats associated with a slight calibrated exercise protocol) MI-RAT OA model.

Comment #2:

            Sample sizes for the allodynia assessment, neuropeptide quantification and histological assessments are not indicated. I assume these were n=12 per group, but it is conceivable that sample sizes were smaller for the molecular and histological analyses. Please mention sample sizes in the figure legends.

Response no.2:

Sample sizes have been added in accordance with the request of Reviewer #2 in each figure legends.

Comment #3:

            The symbols indicating between-group differences in Table 1 are very difficult to interpret. I assume that a,b,c,d refers to the experimental groups 2-4 in the Table? Please indicate in the legend what between-group difference is meant here and check if these are correct. 

Response no.3:

The authors thank the reviewer for this appropriate comment. As indicated in the answer no. 3 to the Reviewer #1, we have reviewed and revised the annotations in Table 1 to emphasize statistical significance of the spinal neuropeptide concentrations. This involved adding footnotes with specific explanations (lines 168-172) for each group comparison.

Comment #4:

            I am not sure the statement in line 178-180 is correct. None of the naive-OVX values seem to differ significantly from sedentary-PA (letter c) or exercise-PA (letter d). I understand it is tempting to explain the improved pain outcome by neuropeptide levels returning to baseline, but this also happens in sedentary-PA groups. Please rephrase and discuss whether other mechanisms (muscular?) may have contributed to improved pain outcomes.

Response no.4:

To address the potential for misinterpretation of the various group comparisons of neuropeptide concentrations, as mentioned in the previous response no. 3, we have revised the annotations of Table 1. Furthermore, we have also included the p-values in the corresponding text (lines 173-192) in the result section 2.2. With these modifications, it becomes more evident that Exercise-PA was the only group to present similar neuropeptide concentrations as Naïve-OVX. Sedentary-PA was significantly different than Naïve-OVX group for CGRP (increased) and Met-ENK (decreased). The data interpretation, as presented in the Discussion, is supported by lot of neuropathological processes:

  • In OVX rats, the surgical model induction (Sedentary-placebo) is associated with an increase in pronociceptive neuropeptides (SP and CGRP), compared to Naïve-OVX.
  • In OVX rats, the addition of exercise to the surgical model (Exercise-Placebo = validated MI-RAT model) corresponds to a maintained increased SP compared to Naïve-OVX, but yes decreased SP compared to Sedentary-Placebo, and a reduction in CGRP, both elements being transduced as a possible anti-pronociceptive effect of exercise. Moreover, Exercise-Placebo was associated with increased BK, Met-ENK, and Leu-ENK compared to both Sedentary-Placebo and Naïve-OVX. For SST, the increase was close to statistical significance. Such increase in SST, Met-ENK and Leu-ENK was associated with stimulated descending nociceptive inhibitory control.
  • Finally, the effect of PA (positive analgesic = multimodal association of pregabalin and carprofen from D3 to D56) translated in a decrease in SP and an increase in BK, for both Sedentary-PA and Exercise-PA compared to Sedentary-Placebo. Therefore, the combination of Exercise-PA was associated with a reduction of both pronociceptive neuropeptides SP (effect of both exercise and PA) and CGRP (effect of exercise). If PA decreased Met-ENK in Sedentary-PA compared to Naïve-OVX, the combination Exercise-PA was associated to combined effect of exercise and PA on BK (increased = PA and/or exercise effect), and decrease in SST, Met-ENK and Leu-ENK compared to Exercise-Placebo (= effect of PA), but the effect on Met-ENK was an higher concentration, compared to Sedentary PA (= effect of exercise). At the end, it is remarkable that the Exercise-PA group was the only one to present neuropeptide concentrations identical to the Naïve-OVX group.

Moreover, section 3.3 (lines 415-417) of the discussion part has been bonified to include additional possible mechanisms that could have contributed to the improved pain outcomes.

Comment #5:

            It is unconventional to express modified Mankin score as percentage of cartilage alterations (Table 2 - line 514 - 528) and I have difficulties interpreting these values. I assume the maximal score of 50 (medial and lateral tibia) would represent 100% cartilage alteration? Authors have not followed the OARSI recommendations (ref 80) as claimed in line 520. I am not aware of validated scoring systems that have 10 subgrades for structural changes, 6 for proteoglycan loss and 6 for chondrocyte cellularity. Please use validated scoring systems to assess histological changes such as PMID: 11924812. Authors should keep in mind that histological scores are to be treated as non-parametric data and using percentages is not meaningful. Between-group differences should be evaluated using non-parametric testing on scores, rather than percentages.

Response no.5:

We thank the reviewer for their thorough and detailed commentary to improve the presentation of histological results.

As mentioned in the second response of the Reviewer #1, we have incorporated additional information (lines 208-221; 556-558; 565-566; 576-578) to enhance the understanding of the results, their interpretation, the rigor applied in the evaluation of the stifle joints and the validity of the scoring table. The complete and detailed modified Mankin’s scoring table (adapted from [7-9]) used to assess the cartilage alteration of the stifle in Table A1 is also now available in Appendix A. The Reviewer #2 is right, the maximum (25 + 25) score is 50 (representing 100% of cartilage alteration), very close to the one (26 per section) cited in Karahan et al., 2001 (PMID: 11924812, [16])

Additionally, we have provided the corresponding references in section 4.2 (line 552) of the revised manuscript. The table A1 already incorporates OARSI histological guidelines published by Gerwin et al., 2010 [9]. Furthermore, one of the blinded observers who scored the stifle was an experienced veterinary pathologist (A.B.) with extensive expertise in histopathological evaluation and a strong familiarity with the use of the modified Mankin’s scoring table adapted from [7-9]. The modified Mankin’s score is recognised as focusing specifically on cartilaginous damages when the OARSI scale is used for whole joint alterations. In this validation step of our model, we did hypothesize the cartilage to be the most demonstrative section for induced lesions, as other joint alterations (sclerosis, osteophytosis, etc.) to occur later. Finally, Table 2 in section 2.3 have been also enhanced to improve the visual presentation of the histological results.

Finally, inferential analysis has been redone by using the histological scores with a non-parametric test. The results maintain the same significant differences, with updated P-values that are more precise in this revised manuscript version.

Comment #6:

The discussion should be shortened where possible. Authors should refrain from summarizing results (e.g. line 343-355). I suggest focusing the discussion on the following aspects i) What are the advantages of the novel model compared to other experimental OA rat models? ii) What factors other than neuropeptides may have contributed to better pain outcomes by combined treatment and iii) What is the potential translational value of the model, a specific OA population perhaps?

Response no.6:

The suggestions made by the Reviewer #2 for the different elements of discussion are relevant, and still included, but simplified and integrated differently in the manuscript. Indeed, the advantages (section 3.3 of discussion part) of the updated MI-RAT model focused on the differences compared to our first publication by Gervais et al., 2019 [10]. To avoid excessive redundancy, they are summarized, particularly the comparison with the chemical model, in the discussion (lines 236-250). Moreover, section 3.3 includes further advantages (lines 379-389), followed by several components highlighting the translational value of the model. We are not sure about the Reviewer #2’s comment about summarizing results (e.g. line 343-355). In our interpretation, this section (line 352-360) presents a mechanistic hypothesis about the decrease in chondrocyte loss observed with exercise. This is supported by corresponding references. The second part (line 361-364) highlights the factual evidence of absent supplemental structural alteration that could have been expected with PA use. Finally, beyond neuropeptide quantification, the utilization of static quantitative sensory testing such as PWT for detecting pain modulation is a reliable and sensitive outcome measure for assessing different conditions (exercise, treatment) in the MI-RAT model. This is discussed in Sections 3.1 (lines 264-273) and 3.2 (lines 365-372) of the manuscript. We agree with the Reviewer #2 that other factors than neuropeptide spinal concentration could explain the observed results, e.g. epigenomic modulation. However, this is a topic of future investigation for our group, and as highlighted by the reviewer, the Discussion is long enough (and we didn’t provide data to support it). In our hands, spinal concentration of neuropeptides has been a highly sensitive and specific marker of induced pain, and responsiveness to various analgesics both in chemical (MIA) or surgical (MI-RAT) OA rodent models. In this study, the fact that all tested neuropeptides concentration in the Exercise-PA group was identical to this of the Naïve-OVX group is a remarkable result, supporting the predictive validity of the model (Line 409-417). Finally, we considered to have indicated in several places in the manuscript that the MI-RAT model is particularly representative of the post-traumatic OA population.

Comment #7:

I hope my comments and suggestions are deemed useful.   

Response no.7:

The authors greatly appreciate Reviewer #2's comments and suggestions, which significantly contributed to the manuscript's improvement. Sincere thanks to Reviewer #2 for their time and effort in reviewing our work.

Comment #8:

Several sentences are overly wordy (e.g. lines 53, 55, 55-60, 63-66, 172-176, 197-202). The overall readability of the manuscript could be improved by shortening sentences. AI-writing tools such as ChatGPT could be helpful to correct both grammar and readability using the prompt "Check grammar and improve readability". An AI-based grammar check may also help with incorrect translations from French into English.

Response no.8:

We thank the Reviewer for the suggestion to review the English. The manuscript text has been revised (grammar and syntax improvement) with ChatGPT, and the changes are highlighted in yellow (lines 54-59, 64-68, 170-194, 208-224), as well as in many places in the Discussion or the Materials and Methods (line 465-472) to better describe the calibrated exercise protocol qualified as ‘slight’ intensity, or (line 555-567) for the mMs analysis and (line 571-583) to adapt the Statistical analysis to the updated analyses conducted in agreement of the Reviewer #2’s advices.

References

  1. Galois, L.; Etienne, S.; Grossin, L.; Watrin-Pinzano, A.; Cournil-Henrionnet, C.; Loeuille, D.; Netter, P.; Mainard, D.; Gillet, P. Dose-response relationship for exercise on severity of experimental osteoarthritis in rats: a pilot study. Osteoarthritis Cartilage 2004, 12, 779–786.
  2. Iijima, H.; Aoyama, T.; Ito, A.; Yamaguchi, S.; Nagai, M.; Tajino, J.; Zhang, X.; Kuroki, H. Effects of short-term gentle treadmill walking on subchondral bone in a rat model of instability-induced osteoarthritis. Osteoarthritis Cartilage 2015, 23, 1563–1574.
  3. Brito, R.G.; Rasmussen, L.A.; Sluka, K.A. Regular physical activity prevents development of chronic muscle pain through modulation of supraspinal opioid and serotonergic mechanisms. Pain reports 2017, 2, e618.
  4. Ni, G.X.; Lei, L.; Zhou, Y.Z. Intensity-dependent effect of treadmill running on lubricin metabolism of rat articular cartilage. Arthritis Res. Ther. 2012, 14, R256.
  5. Ni, G.X.; Liu, S.Y.; Lei, L.; Li, Z.; Zhou, Y.Z.; Zhan, L.Q. Intensity-dependent effect of treadmill running on knee articular cartilage in a rat model. Biomed. Res. Int. 2013, 2013, 172392.
  6. Poole, D.C.; Copp, S.W.; Colburn, T.D.; Craig, J.C.; Allen, D.L.; Sturek, M.; O'Leary, D.S.; Zucker, I.H.; Musch, T.I. Guidelines for animal exercise and training protocols for cardiovascular studies. Am. J. Physiol. Heart Circ. Physiol. 2020, 318, H1100–H1138.
  7. Carlson, C.S.; Guilak, F.; Vail, T.P.; Gardin, J.F.; Kraus, V.B. Synovial fluid biomarker levels predict articular cartilage damage following complete medial meniscectomy in the canine knee. J. Orthop. Res. 2002, 20, 92–100.
  8. Colombo, C.; Butler, M.; Hickman, L.; Selwyn, M.; Chart, J.; Steinetz, B. A new model of osteoarthritis in rabbits. II. Evaluation of anti-osteoarthritic effects of selected antirheumatic drugs administered systemically. Arthritis Rheum. 1983, 26, 1132–1139.
  9. Gerwin, N.; Bendele, A.M.; Glasson, S.; Carlson, C.S. The OARSI histopathology initiative - recommendations for histological assessments of osteoarthritis in the rat. Osteoarthritis Cartilage 2010, 18 Suppl 3, S24–34.
  10. Gervais, J.A.; Otis, C.; Lussier, B.; Guillot, M.; Martel-Pelletier, J.; Pelletier, J.P.; Beaudry, F.; Troncy, E. Osteoarthritic pain model influences functional outcomes and spinal neuropeptidomics: A pilot study in female rats. Can. J. Vet. Res. 2019, 83, 133–141.
  11. Keita-Alassane, S.; Otis, C.; Bouet, E.; Guillot, M.; Frezier, M.; Delsart, A.; Moreau, M.; Bedard, A.; Gaumond, I.; Pelletier, J.P.; et al. Estrogenic impregnation alters pain expression: analysis through functional neuropeptidomics in a surgical rat model of osteoarthritis. Naunyn Schmiedebergs Arch. Pharmacol. 2022, 395, 703–715.
  12. Amandusson, A.; Blomqvist, A. Estrogenic influences in pain processing. Front Neuroendocrinol. 2013, 34, 329–349.
  13. Mogil, J.S. Sex differences in pain and pain inhibition: multiple explanations of a controversial phenomenon. Nat. Rev. Neurosci. 2012, 13, 859–866.
  14. Riley, J.L., 3rd; Robinson, M.E.; Wise, E.A.; Price, D. A meta-analytic review of pain perception across the menstrual cycle. Pain 1999, 81, 225–235.
  15. Mogil, J.S. Laboratory environmental factors and pain behavior: the relevance of unknown unknowns to reproducibility and translation. Lab. Anim. N. Y. 2017, 46, 136–141.
  16. Karahan, S.; Kincaid, S.A.; Kammermann, J.R.; Wright, J.C. Evaluation of the rat stifle joint after transection of the cranial cruciate ligament and partial medial meniscectomy. Comp. Med. 2001, 51, 504–512.